# Insights into the ubiquitin transfer cascade catalyzed by the *Legionella* effector SidC

**David Jon Wasilko[1,2], Qingqiu Huang[3], Yuxin Mao[1,2]***

[1]Department of Molecular Biology and Genetics, Cornell University, Ithaca, United States; [2]Weill Institute for Cell and Molecular Biology, Cornell University, Ithaca, United States; [3]MacCHESS, Cornell University, Ithaca, United States

**Abstract** The causative agent of Legionnaires' disease, *Legionella pneumophila*, delivers more than 330 virulent effectors to its host to establish an intracellular membrane-bound organelle called the *Legionella* containing vacuole. Among the army of *Legionella* effectors, SidC and its paralog SdcA have been identified as novel bacterial ubiquitin (Ub) E3 ligases. To gain insight into the molecular mechanism of SidC/SdcA as Ub ligases, we determined the crystal structures of a binary complex of the N-terminal catalytic SNL domain of SdcA with its cognate E2 UbcH5C and a ternary complex consisting of the SNL domain of SidC with the Ub-linked E2 UbcH7. These two structures reveal the molecular determinants governing the Ub transfer cascade catalyzed by SidC. Together, our data support a common mechanism in the Ub transfer cascade in which the donor Ub is immobilized with its C-terminal tail locked in an extended conformation, priming the donor Ub for catalysis.

DOI: https://doi.org/10.7554/eLife.36154.001

## Introduction

*Legionella pneumophila* is an intracellular opportunistic human pathogen that causes a severe form of pneumonia termed Legionnaires' disease (*Cunha et al., 2016*; *Rowbotham, 1980*). Following phagocytic uptake, *Legionella* uses a Dot/Icm (Defective organelle trafficking/Intracellular multiplication) type IV secretion system to secrete over 330 effector proteins into the host cytosol (*Ensminger, 2016*; *Qiu and Luo, 2017*). The concerted action of these effectors leads to the formation of the LCV (*Legionella*-containing vacuole), an organelle that prevents lysosome-mediated degradation of the invading bacteria while also serving as a replicative niche (*Isberg et al., 2009*; *Swanson and Isberg, 1995*; *Tilney et al., 2001*). Although the biological functions of most *Legionella* effectors are as yet unknown, a number of effectors have been shown to modulate a rather diverse array of host cellular processes, including membrane trafficking, cellular signaling, lipid metabolism, and in particular, the host ubiquitination pathway (*Hubber et al., 2013*; *Hubber and Roy, 2010*; *Lin and Machner, 2017*; *Qiu and Luo, 2017*).

Ubiquitination is an essential eukaryotic posttranslational modification involved in myriad cellular activities. Ub is covalently attached to substrates in a three-enzyme cascade (*Hershko and Ciechanover, 1998*). Ub is first activated by an E1 activating enzyme, then transferred to one of approximately 30 E2 conjugating enzymes, and finally ligated to a substrate by one of hundreds of E3 Ub ligases. The E3 ligases, which play a crucial role in the ubiquitination cascade, can be divided into three major classes: Really Interesting New Gene (RING) E3s (*Deshaies and Joazeiro, 2009*), Ring-Between-Ring (RBR) E3s (*Aguilera et al., 2000*), and Homologous to E6AP C-terminus (HECT) E3s (*Huibregtse et al., 1995*; *Rotin and Kumar, 2009*). Each of the three classes of E3s uses a distinct strategy to transfer Ub (*Berndsen and Wolberger, 2014*). The RING E3s promote a 'closed

*For correspondence:
ym253@cornell.edu

**Competing interests:** The authors declare that no competing interests exist.

conformation' of the E2~Ub by locking the C-terminus of Ub into the active site groove on the E2 to facilitate direct Ub transfer from the E2~Ub to a substrate (*Dou et al., 2012*; *Plechanovová et al., 2012*; *Pruneda et al., 2012*). The HECT E3s catalyze two distinct reactions: a transthiolation reaction, which transfers Ub from E2~Ub to the E3 catalytic cysteine residue; and a subsequent aminolysis reaction, which transfers Ub from E3~Ub to a substrate lysine (*Huibregtse et al., 1995*). The third class, the RBR family of E3 ligases, shares features of both RING and HECT E3s and utilizes a hybrid mechanism for Ub transfer (*Dove et al., 2016*; *Wenzel et al., 2011*). Besides the three major classes of E3 ligases found in eukaryotes, a large number of bacterial pathogens encode a variety of effectors that mimic either the RING or HECT type E3 ligases. However, some of these bacterially encoded E3 ligases have little sequence or structural homology to any other E3s (*Ashida et al., 2014*; *Huibregtse and Rohde, 2014*). For example, the *Salmonella* protein SopA and the *Escherichia coli* protein NleL possess E3 ligase activity dependent on a catalytic cysteine and interact with E2s that contain a conserved phenylalanine residue yet show little sequence identity to known HECT E3s (*Kamadurai et al., 2013*; *Lin et al., 2012*). How these unique E3 ligases catalyze Ub conjugation remains elusive.

The *Legionella* effector protein SidC and its paralog SdcA have been shown to be required for the recruitment of ER proteins and ubiquitin signals to the LCV during an infection (*Hsu et al., 2014*). Our previous work has further shown that the SidC and SdcA are bona fide Ub E3 ligases (*Hsu et al., 2014*; *Luo et al., 2015*). SidC contains an N-terminal Ub Ligase (SNL) domain, a P4C domain that specifically binds phosphatidylinositol-4-phosphate (*Luo et al., 2015*; *Ragaz et al., 2008*) and a C-terminal portion of unknown function. Despite the lack of sequence or structural homology of SidC to any known eukaryotic E3s, the SNL domain of SidC uses a conserved cysteine as its catalytic residue, which is reminiscent of the eukaryotic HECT family of E3s. Our previous structural studies revealed that the SNL domain of SidC contains two lobes, a larger main lobe harboring the catalytic cysteine and a smaller E2-binding lobe connected to main lobe via two flexible linker peptides (*Hsu et al., 2014*; *Luo et al., 2015*). However, the molecular mechanism of this unique family of bacterial Ub E3 ligases is largely unknown.

To gain insight into the molecular basis of the ubiquitination reaction catalyzed by SidC/SdcA, we determined crystal structures of two SidC/SdcA complexes: a binary complex of the SNL domain of SdcA with the E2 UbcH5C and a ternary complex of the SNL domain of SidC with the E2 UbcH7 covalently linked to ubiquitin (UbcH7~Ub). The complex structures reveal that upon binding of E2~Ub, the INS lobe undergoes a drastic swiveling to close a nearly 80 Å gap between the two catalytic cysteines of the E2 and E3, respectively. As a consequence, the donor Ub is clamped between the INS and main lobes and makes an extensive network of contacts with the SNL domain. In particular, the C-terminal tail of the donor Ub adopts an extended β conformation and zip-pairs with a short β-strand upstream of the catalytic cysteine of the SNL domain. Mutations that interfere with the interactions between the donor Ub and the E3 ligase impede the ligase activity. Thus, our data not only reveal the catalytic mechanism of a unique family of bacterial E3 ligases, but also underpin a general concept of the ubiquitination reaction that efficient Ub transfer requires the donor Ub to be placed in a stationary position with the C-terminal tail of the donor Ub locked at the catalytic site.

## Results

### Overall structure of the SdcA-UbcH5C complex

As the first step in deciphering the mechanism of the SidC family of Ub E3 ligases, we sought to investigate how SidC recognizes E2. Our previous studies showed that SidC and SdcA preferentially carry out ubiquitination when either UbcH5C (UBE2D3) or UbcH7 (UBE2L3) serves as the E2 (*Hsu et al., 2014*). We screened all combinations of the SNL domain of SidC and SdcA with either UbcH5C or UbcH7 in crystallization trials and successfully obtained crystals formed by the SNL domain of SdcA (a.a. 1–538) and UbcH5C. The crystals diffracted to 3.0 Å and the structure of the binary complex was solved using molecular replacement using our previously determined SidC (a. a.1–542) structure as the search model. The final structural model was refined to an R-factor of 20.3 and a free R-factor of 27.1 (*Table 1*). The SNL domain of SdcA, which shares about 72% sequence homology to SidC, has a bilobed structure containing a large main lobe where the catalytic cysteine (C45) resides and a smaller inserted lobe (INS lobe) (*Figure 1*). We have previously shown that the

**Table 1.** Data collection and refinement statistics.

| | UbcH7 ~ Ub SidC SNL (PDB ID: 6CP2) | UbcH5C-SdcA SNL (PDB ID: 6CP0) |
|---|---|---|
| Data collection | | |
| Space group | P6$_5$22 | C222$_1$ |
| Cell dimensions | | |
| a, b, c (Å) | 101.522, 101.522, 352.302 | 135.550, 142.202, 118.333 |
| a, b, g (°) | 90.0, 90.0, 120.0 | 90.0, 90.0, 90.0 |
| Resolution (Å)* | 50.0–2.90 (2.95–2.90) | 50.00–2.90 (2.95–2.90) |
| $R_{sym}$[†] (%) | 14.0 (80.2) | 12.5 (95.3) |
| I/σ(I) | 30.3 (16.6) | 8.8 (2.7) |
| Completeness (%) | 99.9 (99.9) | 96.4 (91.5) |
| Redundancy | 13.8 (14.6) | 4.0 (3.7) |
| Refinement | | |
| Resolution (Å) | 2.9 | 3.0 |
| No. reflections | 27,387 | 23,102 |
| $R_{work}$/$R_{free}$[‡] (%) | 21.7/28.3 | 20.3/27.1 |
| R.m.s. deviations | | |
| Bond lengths (Å) | 0.0113 | 0.0106 |
| Bond angles (°) | 1.4734 | 1.4619 |
| Ramachandran Plot | | |
| Preferred (%) | 96.84 | 96.94 |
| Allowed (%) | 3.16 | 3.06 |
| Disallowed (%) | 0 | 0 |

*Values in parentheses are for highest-resolution shell.

[†]$R_{sym} = \Sigma_h\Sigma_i|I_i(h) - <I(h)|/\Sigma_h\Sigma_i I_i(h)$.

[‡]$R_{crys} = \Sigma(|F_{obs}| - k|F_{cal}|)/\Sigma|F_{obs}|$. $R_{free}$ was calculated for 5% of reflections randomly excluded from the refinement.

DOI: https://doi.org/10.7554/eLife.36154.003

SNL domain of wild type SidC forms a complex with E2~Ub, but SidC lacking the INS lobe fails to do so (*Luo et al., 2015*). In agreement with this observation, the binary complex structure reveals that the INS lobe indeed mediates direct binding with the E2, UbcH5C. The overall structure of the complex has an elongated arch-like shape. Strikingly, the catalytic cysteine of UbcH5C (C85) is about 80 Å away from the catalytic C45 of the SNL domain (*Figure 1B–E*). This structural feature indicates that a drastic conformational rearrangement is required for the INS lobe to bring the two catalytic cysteines within attacking distance for efficient Ub transfer between the E2 and the E3.

## Overall structure of the SidC-UbcH7 ~ Ub complex

To gain insight into how this conformational rearrangement occurs during catalysis, we sought to determine the structure of the SNL domain in complex with a Ub-charged E2. To overcome the labile nature of a thioester bond, the catalytic cysteines of both UbcH5C and UbcH7 were mutated to lysine to form an E2~Ub complex linked by a stable isopeptide bond. In addition, to enhance the stability of the E3s, the catalytic cysteines of SidC (C46) and SdcA (C45) were mutated to alanine. We then performed crystallization screens for all four possible combinations of SidC or SdcA with UbcH5C~Ub or UbcH7~Ub, and obtained crystals of the SNL domain of SidC with UbcH7~Ub when mixed together in a 1:1.2 molar ratio. Although the crystals diffracted poorly with the conventional flash-frozen cryoprotection method, the crystals diffracted to 2.9 Å when cryo-cooled under high-pressure (*Huang et al., 2016*). The structure of the ternary complex was also solved via molecular replacement using the main lobe of SidC as the search model, and the final structure was refined to an R-factor of 21.7 and a free R-factor of 28.3 (*Table 1*). The SidC-UbcH7~Ub complex has a more globular, compact structure (*Figure 2*). In the ternary complex, the SNL domain of SidC maintains an

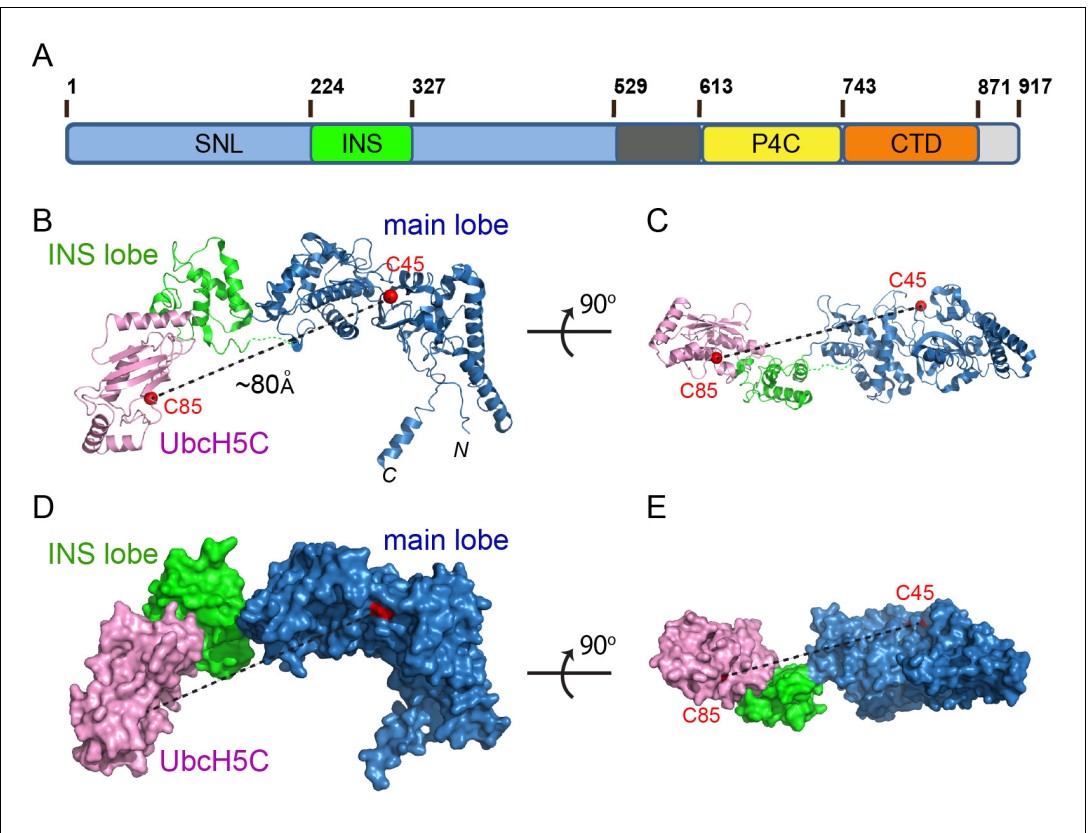

**Figure 1.** Crystal structure of the SdcA-UbcH5C binary complex. (**A**) Domain architecture of a member of the SidC family of Ub E3 ligase. (SNL: SidC N-terminal Ub E3 ligase; INS: insertion lobe; P4C: PI(4)P binding of SidC; CTD: C-terminal domain.) (**B–C**) Two orthogonal views of a ribbon diagram of the SNL domain of SdcA (blue: main lobe; green: INS lobe domain) in complex with UbcH5C (pink). The catalytic cysteines of both SdcA (C45) and UbcH5C are shown as red spheres. Note that the distance between the two cysteines is about ~80 Å. (**D–E**) Two orthogonal views of the SdcA-UbcH5C complex presented in surface.

DOI: https://doi.org/10.7554/eLife.36154.002

arch-like shape, although with a shortened span compared to the SNL domain of SdcA in the binary complex. UbcH7 interacts with both the main and INS lobes of the SNL domain. The surface area of the INS lobe that mediates the binding with UbcH7 is nearly identical to that observed in the SdcA-UbcH5C binary complex. The Ub moiety is tightly clamped between the INS and main lobes, filling the space under the 'arch'. Remarkably, the distance between the two Cα atoms of the catalytic residues of the E2 (C86K) and the E3 (C46A) is reduced to ~9 Å (from ~80 Å in the binary complex) and the C-terminal carbonyl group of the donor Ub is within attacking distance from the modeled sulfhydryl group of C46A on SidC (*Figure 2A–D*). This dramatic structural rearrangement is mainly caused by a swinging motion made by the E2-binding INS lobe.

## Conformational flexibility of the E2-binding INS lobe

The INS lobe is hinge-anchored to the main lobe through two flexible loops (*Figure 1B*). A structural comparison of the SNL domains from the two complexes and two previously reported SidC structures reveals that the INS lobe swings within a range of $90^0$ between the most extended conformation observed in the SdcA-UbcH5C complex and the most compact conformation found in the SidC-UbcH7~Ub complex, with the previously solved structures of SidC apo-enzymes falling in the middle (*Figure 3A–B* and *Figure 3—figure supplement 1*). Large conformational changes have commonly been observed in HECT-type E3 ligases. The classic HECT domain comprises a larger E2-binding N-lobe and a smaller C-lobe that harbors the catalytic cysteine. Structural studies have shown that a flexible hinge-like linker between the two lobes to allows the C-lobe cysteine to approach the

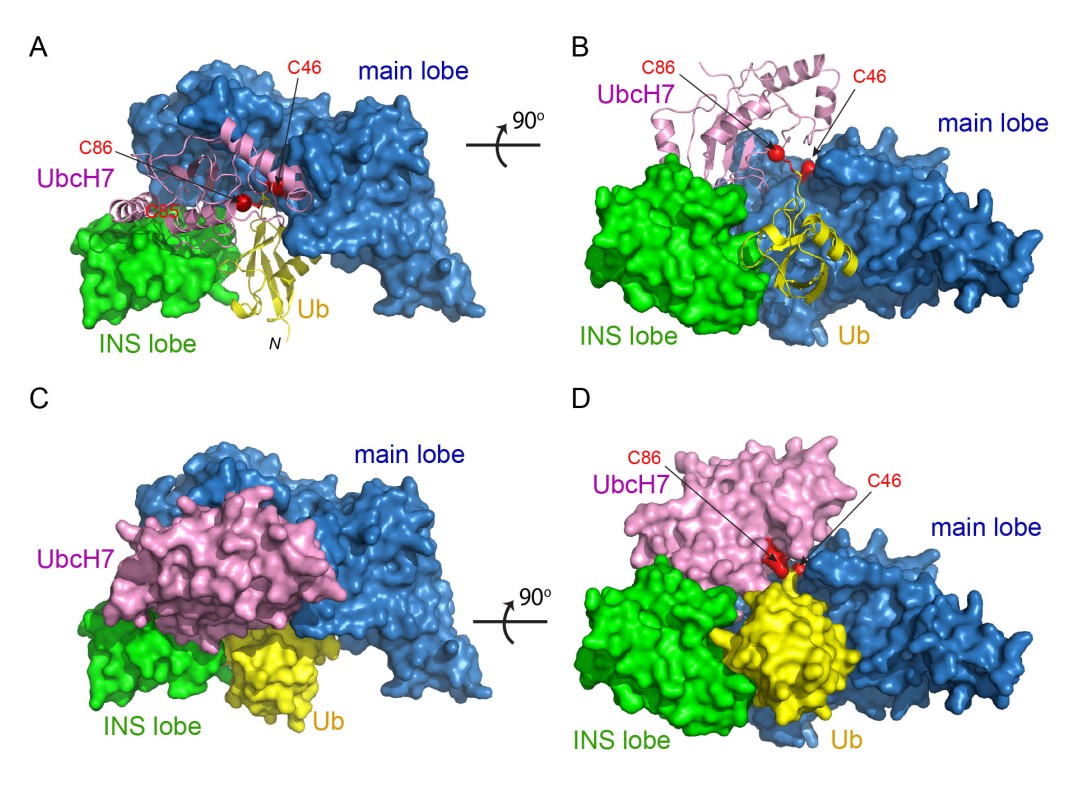

**Figure 2.** Crystal structure of the SidC-UbcH7~Ub ternary complex. (**A–B**) Two orthogonal views of the SidC-UbcH7~Ub complex structure. The SNL domain of SidC is shown in surface with the main lobe colored in blue and INS lobe in green. UbcH7 (pink) and the donor Ub (yellow) are shown in ribbon. The catalytic cysteines of both SidC (C46) and UbcH7 (C86) are shown in red. (**C–D**) Two orthogonal views of the SidC-UbcH7~Ub ternary complex in surface representation.

DOI: https://doi.org/10.7554/eLife.36154.004

E2~Ub thioester for the transthiolation reaction, or move in close proximity to a substrate lysine for the aminolysis reaction in the second step of catalysis (*Huang et al., 1999*; *Kamadurai et al., 2013*; *Kamadurai et al., 2009*; *Verdecia et al., 2003*) (*Figure 3C*). Similar conformational changes were also observed in bacterial HECT-like E3 ligases (*Lin et al., 2012*) (*Figure 3D*). In all known HECT or HECT-like E3s, the catalytic cysteine residing at the edge of the smaller C-lobe cycles between interacting with the E2~Ub thioester and the substrate lysine, owing to the rotational motion of the C-lobe. However, SidC differs in that the larger main lobe containing the catalytic cysteine likely harbors the substrate binding site and thus considered as the stationary lobe relative to the potential substrate binding site, while the E2-binding INS lobe recruits E2~Ub and delivers Ub to the catalytic cysteine through a large swiveling conformational shift. This variation suggests that SidC might use a different mechanism for Ub transfer (discussed below).

## Structural determinants for E2 recognition by SidC

A comparison of the SdcA-UbcH5C interface with that observed in the SidC-UbcH7~Ub complex reveals common structural determinants for E2 recognition. Both SdcA and SidC apply a similar surface area of about 950 Å$^2$ on the INS lobe to bind a region around the conserved E2 loop4 phenylalanine (UbcH5C F62 and UbcH7 F63) (*Figure 4A*), which is the pivotal residue mediating the binding with HECT E3s (*Nuber and Scheffner, 1999*). Further structural analysis reveals two key determinants governing the binding between the INS lobes with E2s. First, the conserved E2 loop4 phenylalanine is placed into a hydrophobic pocket on the INS lobe in both SdcA and SidC (*Figure 4—figure supplement 1*). In SdcA, this hydrophobic pocket is lined with residues I285, A286, and Y273 (*Figure 4B*). Likewise, the hydrophobic pocket on the SidC INS lobe is formed by residues Y292, L297, M298, L299, and A316. Similar to HECT E3s, these residues are not conserved although the

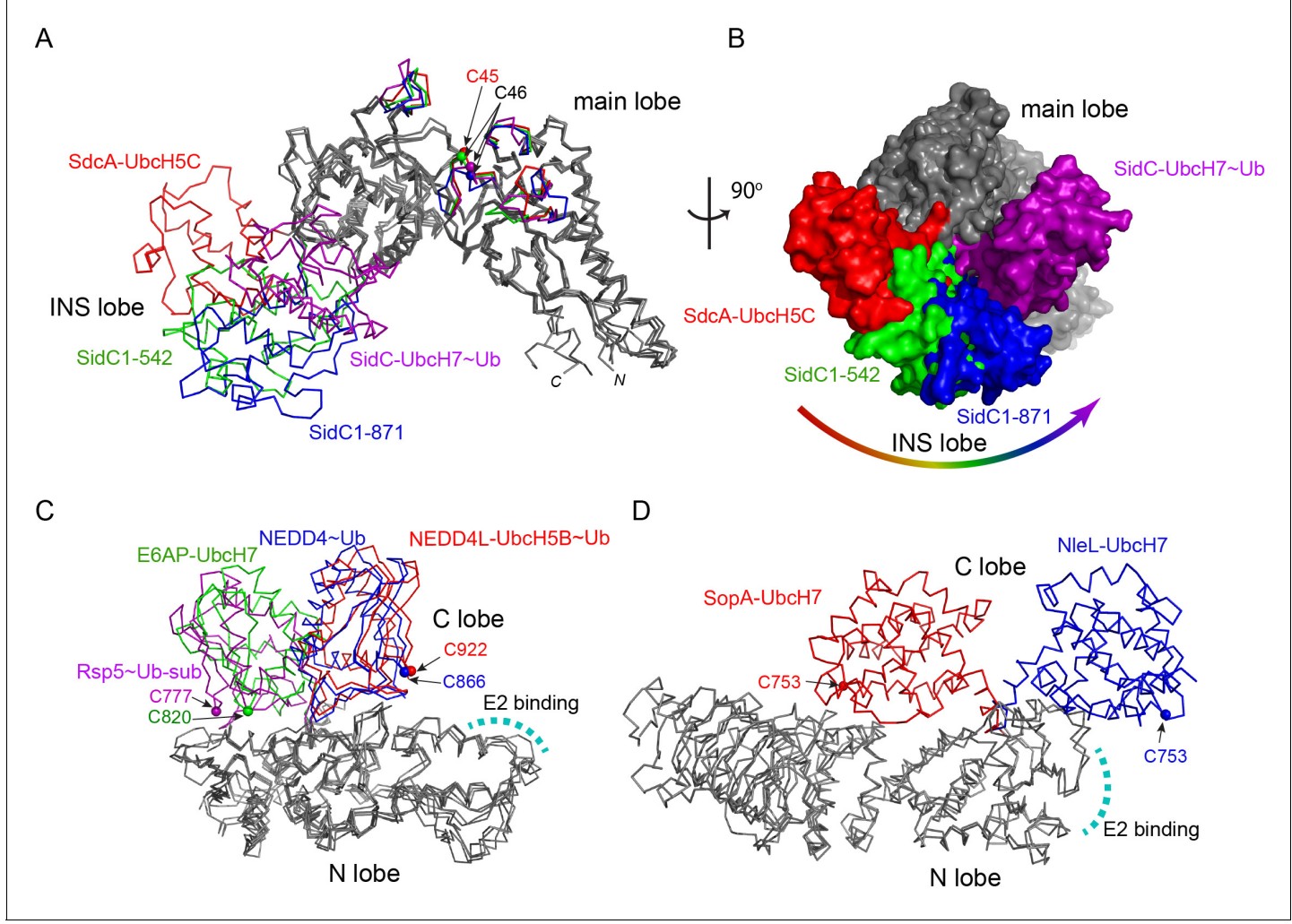

**Figure 3.** Conformational flexibility of the E2-binding INS lobe. (**A**) Overlay of structures of the SNL domains of SidC/SdcA. Four structures were superimposed based on the Cα atoms of the main lobe. The INS lobes adopt a wide range of conformations relative to the main lobe and are colored in red (SdcA-UbcH5C complex), green (SidC 1–542, PDB ID 4TRG), blue (SidC 1–871, PDB ID 4ZUZ), and purple (SidC-UbcH7~Ub), respectively. (**B**) An orthogonal view of the four superimposed structures shown in surface. The INS lobes are colored with the same scheme as in (**A**). (**C**) Structural overlay of HECT domains with their N-lobes superimposed. The C-lobes are colored red (NEDD4L-UbcH5B~Ub, PDB 3JW0), blue (NEDD4~Ub, PDB 4BBN), green (E6AP-UbcH7, PDB 1D5F), and purple (Rsp5~Ub sub, PDB 4LCD), respectively. The E2 binding site is indicated by a curved dashed line. (**D**) Structural overlay of two HECT-like bacterial E3s. The E2-binding N-lobe is superimposed. The mobile C-lobe is colored in red (SopA-UbcH7, PDB 3SY2) and blue (NleL-UbcH7, PDB 3SQV), respectively.

DOI: https://doi.org/10.7554/eLife.36154.005

The following figure supplement is available for figure 3:

**Figure supplement 1.** A zoomed-in view of the flexible connecting loop region between the main lobe and INS lobe.

DOI: https://doi.org/10.7554/eLife.36154.006

hydrophobic nature of the pocket is maintained. Second, the interaction between E2 and SdcA/SidC is also mediated by complementary electrostatic interactions. The interface surrounding the loop4 phenylalanine on both UbcH5C and UbcH7 has a positive electrostatic potential, while the corresponding interface on the INS lobe of both SdcA and SidC is negatively charged (*Figure 4—figure supplement 2*). In the SdcA-UbcH5C complex, residues K4 and K63 of UbcH5C form salt bridges with SdcA residues D287 and D305. Moreover, the R5 side chain of UbcH5C H-bonds with the main chain carbonyl group of A286 of SdcA (*Figure 4B*). Similarly, in the SidC-UbcH7~Ub complex, K9 of UbcH7 salt bridges with SidC D301 and R6 and K100 of UbcH7 form H-bonds with the carbonyl group of SidC L297 and L299, respectively (*Figure 4C*).

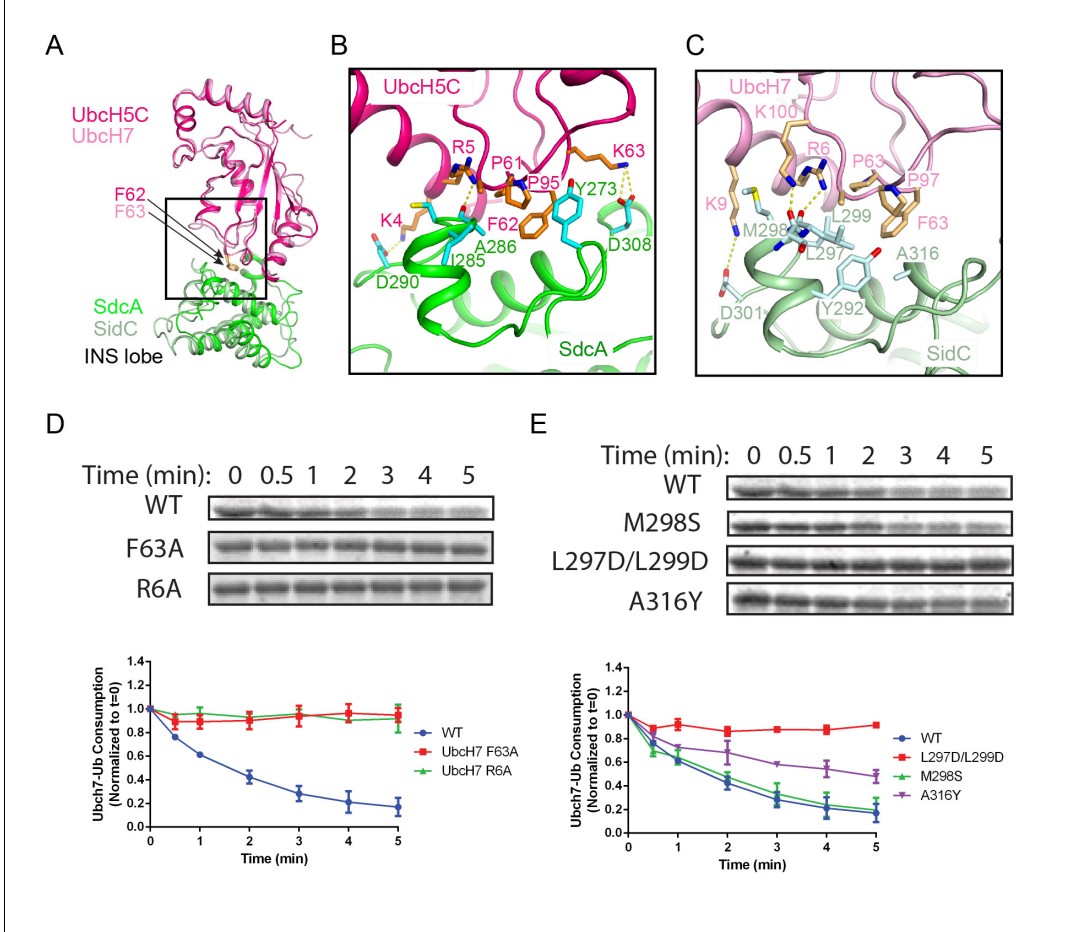

**Figure 4.** E2 recognition by the INS lobe of SidC/SdcA. (**A**) Structural superposition of the INS lobe of SdcA (green) and its bound E2 UbcH5C (pink) with the INS lobe of SidC (light green) and its bound UbcH7 (light pink). (**B**) Zoomed-in view of SdcA/UbcH5C interface. (**C**) Detailed interactions at the interface of SidC/UbcH7. (**D**) Single turnover activity assays of UbcH7 mutants at the SidC/UbcH7 interface. Top panel: representative SDS-gel of UbcH7~Ub complex remaining at the indicated time points. Bottom panel: Quantified intensity of UbcH7~Ub at the indicated time points. The error bar represents the standard deviation of three independent experiments. (**E**) Single turnover activity assays of SidC mutants at its E2 binding interface.
DOI: https://doi.org/10.7554/eLife.36154.007

The following figure supplements are available for figure 4:

**Figure supplement 1.** Surface hydrophobicity analysis of the INS lobes.
DOI: https://doi.org/10.7554/eLife.36154.008

**Figure supplement 2.** Analysis of the surface electrostatic potential at the interface between the E2 and the INS lobe.
DOI: https://doi.org/10.7554/eLife.36154.009

**Figure supplement 3.** Single turnover activity assays of SdcA and UbcH5C mutants at the SdcA/UbcH5C interface.
DOI: https://doi.org/10.7554/eLife.36154.010

To validate the importance of the two types of interactions at the E2-E3 interface, we selectively mutagenized residues at the SidC-UbcH7 interface and analyzed the enzymatic activities of those mutants. Owing to the rapid transthiolation reaction catalyzed by SidC/SdcA, we used the ester-linked UbcH7~Ub (UbcH7 C86S was precharged with Ub by E1) in experiments to measure the single-turnover rate of UbcH7~Ub. The activity of the F63A mutant was substantially reduced compared to wild type (*Figure 4D*), suggesting that the hydrophobic loop4 phenylalanine is indispensable for E2 recognition by SidC/SdcA. Moreover, the R6A mutant, designed to disrupt electrostatic and H-bonding interactions with SidC, also showed a severe impairment of activity. On the other hand, mutations of the SNL domain of SidC also displayed variable effects on the enzymatic activity. A disruption of the hydrophobic pocket by the L297D/L299D double mutant substantially impaired the activity while the single M298S mutant showed no obvious effect. Interestingly, the A316Y mutant,

which was expected to maintain the hydrophobic nature of the pocket but with a reduced pocket size due to its bulky side chain, displayed a moderate effect (*Figure 4E*). The hydrophobic interaction appears to be conserved between SdcA and its cognate E2 UbcH5C. The Ub transfer activity was substantially reduced when the loop4 phenylalanine F62 in UbcH5C was mutated to alanine or when the hydrophobic pocket in the INS lobe of SdcA was disrupted by the A286Y mutant (*Figure 4—figure supplement 3*). Together, our data support that both the hydrophobic and electrostatic/H-bond interactions at the E2-INS interface are crucial for E2 recognition and thus the catalytic function of SidC/SdcA.

## The C-terminal tail of the donor Ub is locked at the E3 catalytic site

The SidC-UbcH7~Ub structure further reveals how the donor Ub is positioned on the SidC SNL domain and primed for transfer. The donor Ub, which is covalently attached to UbcH7, extends away from the E2 and is trapped between the INS and the main lobes of the SNL domain (*Figure 5A*). Notably, the C-terminal tail of the donor Ub adopts a β-strand conformation and pairs in parallel with a short β-strand upstream of the catalytic cysteine. This β-sheet augmentation interaction was also observed in the donor Ub that was covalently attached to the HECT domain of Nedd4 (*Maspero et al., 2013*). The interaction between the donor Ub C-terminal tail and the SNL domain is further reinforced by sidechain-mediated H-bonds and salt bridges. The hydroxyl group of SidC T45 H-bonds with the amino group of Ub G75. Strikingly, R74 of the donor Ub forms a bidentate salt bridge with the highly conserved SidC D43 and a salt bridge with another conserved aspartic residue D51. Disruption of these interactions substantially hampers SidC activity as evidenced by the loss of activity of the D43R mutant, which disrupts the bidentate salt bridge, as well as the T45V

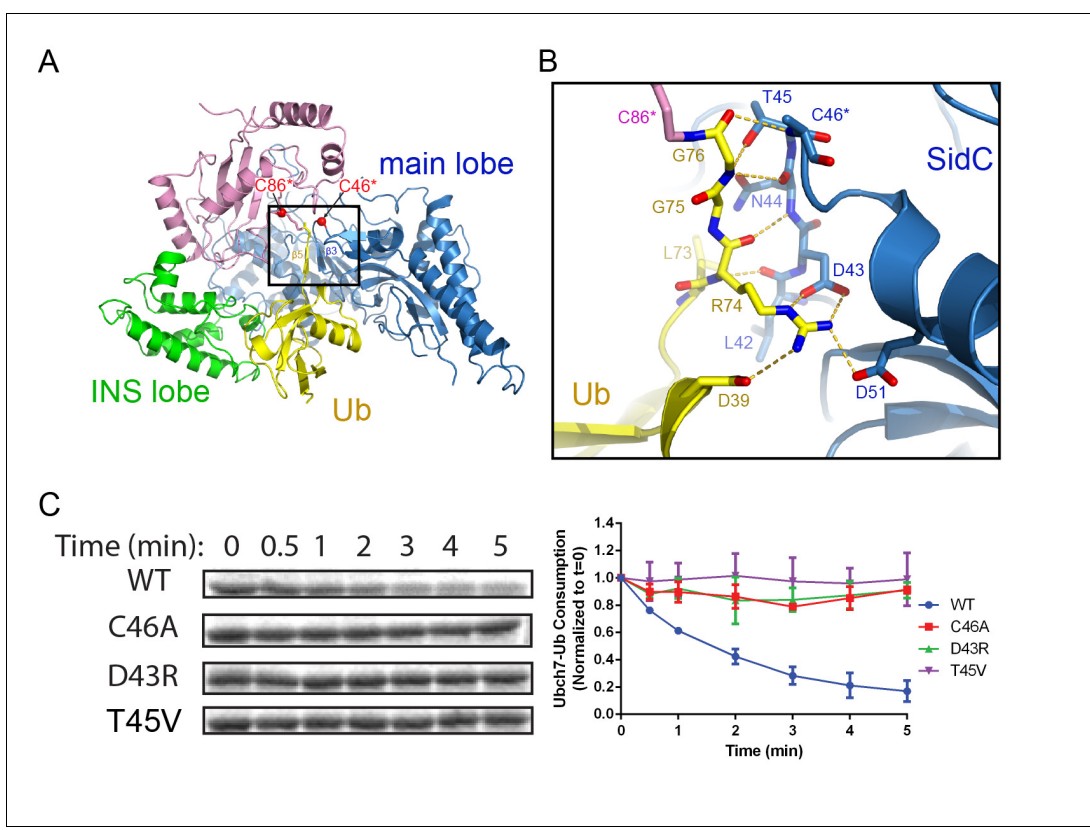

**Figure 5.** The C-terminal tail of the donor Ub is locked on the main lobe. (**A**) Overall view of the interaction of the donor Ub with the SNL domain of SidC. (**B**) Detailed interactions of the C-terminal tail of the donor Ub at the main lobe active site. (**C**) Single turnover activity assays of SidC mutants of residues that interact with the Ub C-terminal tail.

DOI: https://doi.org/10.7554/eLife.36154.011

mutant, which eliminates the hydrogen bond (*Figure 5C*). These results suggest that locking the C-terminal tail of the donor Ub at the E3 catalytic site is crucial for the reaction.

## The donor Ub is held stationary between the INS and main lobes during catalysis

Besides the locked C-terminal tail, the donor Ub also makes extensive contacts with both the INS lobe and main lobe of the SNL domain by burying ~2900 Å$^2$ of surface area (*Figure 6A*). Surface complementarity plays a significant role at the interface between the Ub and the INS lobe. The Ub loop containing K48 protrudes into a shallow groove on the INS domain with the hydrophobic I44 patch embedded at the interface. Surrounding the K48 loop, a network of H-bonds and electrostatic interactions are formed at the interface (*Figure 6B*). In particular, the carbonyl and ε-amino group of K48 H-bonds with the main chain amino and carbonyl groups on the INS lobe, respectively. Furthermore, K6 and R42 form electrostatic interactions with D271 and D319 of the INS lobe. The donor Ub also makes extensive contacts with the main lobe (*Figure 6A*). In addition to the C-terminal tail, the carboxyl end of the first α-helix of Ub forms hydrogen bonds and salt bridges with H504 and K505 on the main lobe (*Figure 6B*). To assess the importance of contacts between the SNL domain and the donor Ub, several residues on the SNL domain that make electrostatic interactions with Ub were mutated to residues carrying opposite charges. Both the D271R and the D319R mutants on the INS lobe largely impair SidC activity, as does the H504D/K505D double mutant on the main lobe. A sequence and structural comparison of SidC with SdcA indicates that residues involved in Ub binding are conserved in SdcA (*Figure 6—figure supplement 1*). Indeed, the Ub transfer activity was also substantially impaired when mutations of these conserved Ub-binding residues were introduced in

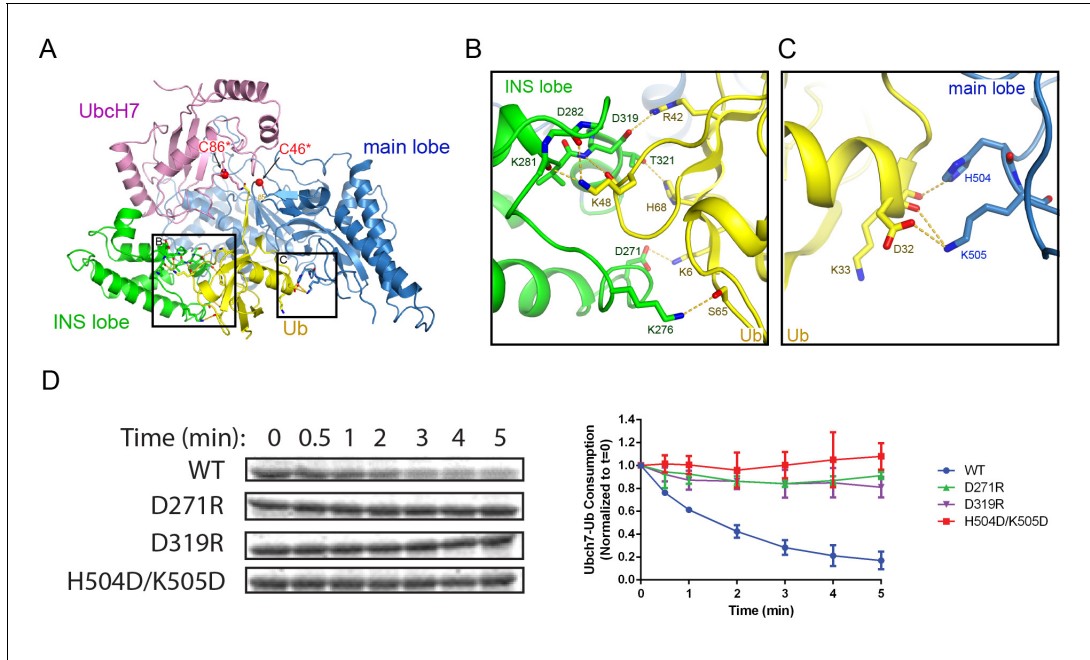

**Figure 6.** The donor Ub is tightly bound between the INS and main lobes of the SNL domain. (**A**) Overall view of the donor Ub bound on the SNL domain of SidC. (**B**) Detailed network of interactions between the donor Ub and the INS lobe. H-bonds are illustrated with dashed lines. (**C**) Zoomed-in view of the interaction between the main lobe and the carboxyl end of the first α-helix of Ub. (**D**) Single turnover activity assays of SidC mutants of selected residues at the interface between the donor Ub and the SNL domain.

DOI: https://doi.org/10.7554/eLife.36154.012

The following figure supplements are available for figure 6:

**Figure supplement 1.** Sequence alignment of the SNL domain region of SidC and SdcA.

DOI: https://doi.org/10.7554/eLife.36154.013

**Figure supplement 2.** Single turnover activity assays of SdcA mutants of selected residues at the interface between the donor Ub and the SNL domain.

DOI: https://doi.org/10.7554/eLife.36154.014

SdcA (*Figure 6—figure supplement 2*). Taken together, these results indicate that the tight binding of the donor Ub by both SNL domain lobes is required for efficient catalysis and may also help prevent the reverse transthiolation reaction (handing the donor Ub back to the E2).

## Acidic residues near the catalytic cysteine are crucial for E3 activity

A structure-based conservation analysis of the SNL domain with the ConSurf server (*Ashkenazy et al., 2016*) revealed a surface patch around the catalytic site enriched with conserved residues (*Figure 7A*). Besides the residues (N44, T45, and C46) comprising the catalytic motif, two invariable aspartic residues (D168, D446) and a histidine residue (H444) constitute a major part of the conserved patch (*Figure 7B*). Although in the SidC-UbcH7~Ub ternary complex the distance of these residues from the catalytic cysteine ranges between 10 and 14 Å, due to the flexible nature of the catalytic loop, these acidic residues can be in close proximity, about 5 Å away from the catalytic cysteine as observed in the SdcA-UbcH5C binary complex (*Figure 7—figure supplement 1*). It has frequently been observed that acidic residues are present in the vicinity the catalytic cysteine in

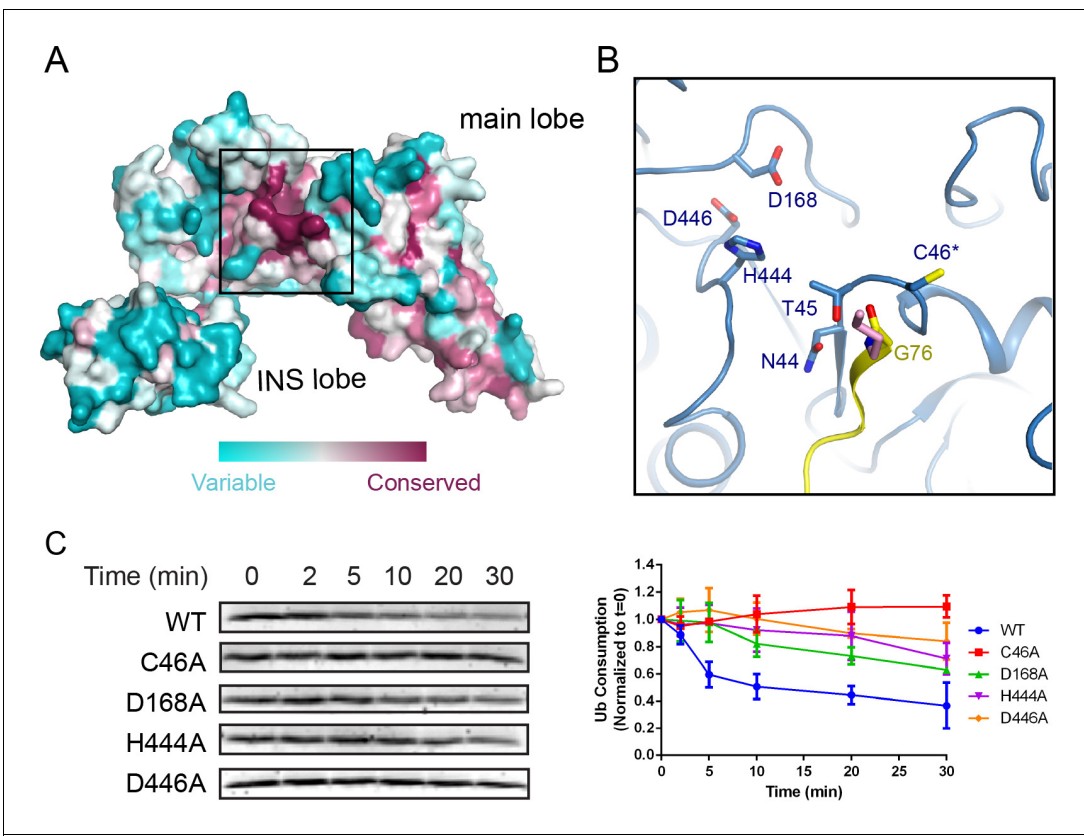

**Figure 7.** Conserved acidic residues near the SNL domain catalytic site. (**A**) Surface conservation analysis of the SNL domain. The conservation was calculated from all SidC homologous sequences from all *Legionella* species with available genomic sequences using the ConSurf server, with the most conserved residues colored in purple and the least conserved residues in cyan. Note that the catalytic site of the SNL domain is concentrated with the most conserved residues. (**B**) A zoomed-in view of the most conserved residues at the catalytic site, including H444 and two acidic residues, D168 and D446. (**C**) Multiple turnover ubiquitination activity assays of SidC mutants of the conserved residues near the SidC active site monitored by the consumption of Ub. Left panel: representative SDS-gel of Ub remaining by the ubiquitination reaction at the indicated time points. Right panel: Quantified intensity of the remaining Ub at the indicated time points. The error bar represents the standard deviation of three independent experiments.

DOI: https://doi.org/10.7554/eLife.36154.015

The following figure supplement is available for figure 7:

**Figure supplement 1.** Conformational flexibility of the SidC and SdcA active sites.

DOI: https://doi.org/10.7554/eLife.36154.016

HECT or HECT-like E3s. These acidic residues were proposed to either guide the substrate lysine into the active site and/or to deprotonate the ε-amino group of the approaching substrate lysine in Ub ligation (*Kamadurai et al., 2013*; *Lin et al., 2012*). In particular, residue H887 in the RBR family ligase, HOIP, acts as a general base to activate the incoming α-amino group for linear Ub chain formation (*Stieglitz et al., 2013*). Similar acidic residues, such as D117 in UbcH5B and D127 in Ubc9, were also found in the vicinity of the catalytic cysteine in E2s (*Berndsen et al., 2013*; *Dou et al., 2012*; *Plechanovová et al., 2012*; *Yunus and Lima, 2006*). In the case of SidC, D168 may function as a guide or directly as a general base for the deprotonation of the attacking lysine. Although slightly further away, D446 forms a hydrogen bond with H444 and thus may indirectly activate the attacking lysine through the deprotonation of H444. Indeed, single amino acid substitutions of either of these three conserved residues markedly impaired the ligase activity of the SNL domain (*Figure 7C and D*). Together, our data reveal an important role for a cluster of invariable acidic residues at the active site in SidC catalyzed ubiquitination.

## Discussion

In this study we present the crystal structures of two homologous *Legionella* Ub E3 ligases SidC/SdcA in complex with a human E2 enzyme alone and an E2 covalently conjugated to ubiquitin. These two structures capture SidC in the stage of E2 recognition, and the following stage at which the donor Ub is positioned for transfer from the E2 to the E3. These two catalytic intermediate snapshots allow us to postulate the mechanism of SidC-catalyzed ubiquitination. Although SidC has no sequence or structural homology to any other E3s, the catalytic SNL domain of SidC contains two lobes, a smaller INS lobe that mediates the binding with E2 and a larger main lobe that carries the catalytic cysteine. The E2-binding INS lobe undergoes a large swinging movement around two hinge peptides that connect the INS lobe to the main lobe. Through the swiveling motion of the INS lobe, E2~Ub is brought into the vicinity of the catalytic site on the main lobe. The donor Ub makes extensive contacts with both the INS and the main lobe and is tightly 'trapped' on the SNL domain with its C-terminal tail engaged in a β-strand augmentation interaction with a β-strand upstream of the catalytic cysteine. Since the catalytic cysteine is at a fixed a position at the center of the catalytic site, it is likely that the donor Ub will maintain the same tightly bound state after the transthiolation reaction, ready for ligation with a substrate lysine during the second step. Several acidic residues neighboring the catalytic site may function to guide and/or activate attacking lysine residues to facilitate Ub ligation.

The structural organization of SidC is reminiscent of the N- and C-lobes in HECT E3s. However, notable differences exist between SidC and HECT E3s. First, in contrast to SidC, the E2-binding N-lobe of HECT E3s is relatively stationary in reference to the substrate binding site, while the catalytic cysteine-bearing C-lobe rotates around the linking peptide between the two lobes to relay the donor Ub from the E2 to a substrate. Second, the donor Ub on HECT E3s makes contacts nearly exclusively with the C-lobe while the donor Ub is clamped between the INS and main lobes and buries much larger surface areas on both the donor Ub and the SNL domain. The third prominent difference is that unlike the stationary donor Ub on SidC, the donor Ub loaded on the C-lobe of HECT E3s undergoes another drastic rotational movement to approach the substrate lysine after the E3~Ub intermediate is formed (*Kamadurai et al., 2013*).

Despite these notable mechanistic differences, our studies of this unique family of bacterial E3s also provide exemplary insights into the general theme of ubiquitination reactions. It has been commonly observed that immobilizing the Ub thioester can enhance the efficiency of Ub transfer (*Dou et al., 2012*; *Maspero et al., 2013*; *Plechanovová et al., 2012*). In the structure of the dimeric RING domain of RNF4 in complex with UbcH5A~Ub, the donor Ub is stabilized by interactions with the both RING protomers and the carboxyl terminus of the donor Ub is locked into an active site groove on the E2 (*Plechanovová et al., 2012*). In the structure that mimics the HECT~Ub thioester intermediate (*Maspero et al., 2013*) and the structure of the HECT domain of Rsp5 crosslinked with a donor Ub and a peptide substrate (*Kamadurai et al., 2013*), Both of the donor Ubs bind to the C-lobe in a similar fashion with their C-terminal tails locked in an extended conformation. The structure of SidC-UbcH7~Ub provides an outstanding example of how the donor Ub is tightly locked on the E3 catalytic domain throughout the two-step ubiquitination reaction. Our results support a

unifying concept that a reduction of the inherent conformational flexibility of the donor Ub thioester on both E2 and E3 is essential for efficient Ub transfer in the ubiquitination cascade.

## Materials and methods

### Cloning and mutagenesis

SidC and SdcA were prepared as previously described (Hsu et al., 2014). In vitro site-directed mutagenesis was used to generate single or multiple point mutations in UbcH5C, UbcH7, SdcA538, and Sid542 using complimentary oligonucleotide primer pairs containing the requisite base substitutions. His-TEV-Ub was generated by PCR amplification of Ub with flanking BamH1 and Xho1 sites. The PCR product was digested and inserted into a pET-His-TEV vector containing an N-terminal 6x-His tag upstream of a TEV cleavage site.

### Protein expression and purification

*E. coli* Rosetta strains harboring the appropriate expression plasmids were grown in Luria-Bertani medium supplemented with 34 µg/ml chloramphenicol and either 50 µg/ml kanamycin or 100 µg/ml ampicillin, and grown to mid-log phase. Protein expression was induced overnight at 18°C with 0.2 mM isopropyl β-D-1-thiogalactopyranoside (IPTG). Harvested cells were resuspended in buffer containing 150 mM NaCl, 50 mM Tris-HCl pH 8.0, and 1 mM PMSF, and lysed by sonication. The lysate was clarified by centrifugation at 31,000 x g for 45 min at 4°C, and incubated with cobalt resin (Gold-Bio) for 1.5 hr at 4°C. Bound proteins were washed extensively with lysis buffer. UbcH5C, UbcH7, SdcA538, and SidC542 were incubated with the SUMO-specific protease Ulp1, to release the desired protein and leave the His-SUMO tag conjugated to the resin. The eluted protein samples were further purified using size-exclusion chromatography (HiLoad Superdex 16/600 S200, GE Lifesciences), and assessed for purity via SDS-PAGE. The fractions corresponding to purified protein were collected, pooled, and concentrated in 150 mM NaCl, 20 mM Tris-HCl, pH 7.5. His-TEV-Ub was bound to cobalt resin, washed extensively, and eluted in 300 mM imidazole, 150 mM NaCl, and 50 mM Tris-HCl pH 8.0, before proceeding to size exclusion chromatography.

Human E1 was purified by reacting with ubiquitin-conjugated to Affi-Gel 10 (Bio-Rad) resin. Briefly, His-TEV-Ub K6 (containing only Lys6, with all other lysines mutated to arginine) was conjugated to the Affi-Gel resin via the primary amine of K6 in 150 mM NaCl, 50 mM HEPES pH 8.0, and then equilibrated in 50 mM Tris-HCl pH 8.0. After lysate clarification, 200 mM ATP, 6 mM creatine phosphate, and 50 U creatine phosphokinase was added to the hE1 supernatant, which was incubated with the Ub-Affi-Gel resin and nutated for 1 hr at 4°C. The mixture was poured into a column and after flow-through collection, the column was washed with 20 column volumes of 0.5 M KCl, 50 mM Tris-HCl pH 8.0. hE1 was eluted in 10 mM DTT, 50 mM Tris-HCl pH 8.0, and a buffer exchange into 150 mM NaCl, 50 mM Tris-HCl pH 8.0 was performed using a centrifugal filter before proceeding to further size exclusion chromatography purification.

### In vitro ubiquitination assays

For single turnover assays, UbcH7 C86S was used to generate stable, ester-linked UbcH7~Ub. 2 µM hE1, 34 µM UbcH7 C86S, and 203 µM ubiquitin were incubated in 50 mM Tris-HCl pH 8.0, 5 mM MgCl$_2$, 1.8 U inorganic pyrophosphatase, 1.8 U creatine phosphokinase, and 21 mM creatine phosphate at 37°C for 8 hr. After the reaction, a centrifugal buffer-exchange was performed (to remove Mg$^{+2}$ and ATP) and a final concentration of 2 mM EDTA was added to chelate any remaining Mg$^{+2}$. The single turnover reaction was performed at room temperature in a total volume of 100 µl. In the reacrtion, 8 µM of ester-linked UbcH7~Ub was mixed with 0.9 µM SidC and 80 µM UbcH5C was added to the reaction, to act as a substrate proxy for SidC activity. The reaction mixture was sampled at the indicated time points and quenched with SDS loading buffer containing 5 mM β-mercaptoethanol. The samples were separated by SDS-PAGE and stained with Coomassie blue. A LI-COR Odyssey scanner was used to image the Coomassie-stained gels, and the ImageStudio software package (LI-COR) was used to quantify the bands. The results were averaged from three independent experiments, and plotted using Graphpad Prism (Graphpad).

Since there are no confirmed substrates that can be specifically ubiquitinated by SidC, in our single turnover assays we used UbcH5C as a substrate proxy. In fact, in our in vitro ubiquitination

reaction, E2s were observed to be ubiquitinated by SidC. The time-dependent accumulation of mono-ubiquitinated UbcH7 migrated at the same size as the ester-linked UbcH7~Ub on an SDS gel. Thus the turnover of ester-linked UbcH7~Ub cannot be distinguished on an SDS gel. By adding an excessive amount of UbcH5C as a substrate proxy, it can prevent/compete SidC mediated ubiquitination of discharged UbcH7. Moreover, since the mono-ubiquitinated UbcH5C migrated at a different size compared to the ester-linked UbcH7~Ub, the consumption of ester-linked UbcH7~Ub can be visualized by SDS-PAGE in a time-dependent manner. The single turnover assay for SdcA was carried out similarly as described above. However, ester-linked UbcH5C~Ub was used in the assay and UbcH7 was used as the substrate proxy. The ubiquitin consumption assays were performed using 2 μM hE1, 3.4 μM UbcH7, 37.5 μM Ub, and 5.1 μM SidC542 (WT or catalytic site mutant) in Tris-HCl pH 8.0, $MgCl_2$, 1.8 U inorganic pyrophosphatase (Sigma-Aldrich), 1.8 U creatine phosphokinase (Sigma-Aldrich), and 21 mM creatine phosphate (Sigma-Aldrich) at 37°C for the indicated timepoints. Samples were analyzed by SDS-PAGE and quantification of the reaction was carried out similarly as described above.

## SdcA/UbcH5C and SidC/UbcH7~Ub complex formation

To form the binary complex, SdcA and UbcH5C were mixed in a 1:1.2 molar ratio, incubated for 1 hr at 4°C, and concentrated to a final SdcA concentration of 10 mg/ml. To form the SidC-UbcH7~Ub ternary complex, UbcH7~Ub was first generated using an in vitro ubiquitination reaction. 2 μM hE1, 34 μM UbcH7 C86K, and 203 μM His-TEV-ubiquitin were incubated in 50 mM Tris-HCl pH 8.0, 5 mM $MgCl_2$, 1.8 U inorganic pyrophosphatase, 1.8 U creatine phosphokinase, and 21 mM creatine phosphate at 37°C for 24 hr. The reaction was mixed with cobalt resin (Gold-Bio) equilibrated in 150 mM NaCl, 50 mM Tris-HCl pH 8.0 and rotated for 2 hr at 4°C. After extensive washing with 150 mM NaCl, 50 mM Tris-HCl pH 8.0 (to remove unreacted UbcH7), bound proteins were eluted in 300 mM imidazole, 150 mM NaCl, 50 mM Tris-HCl pH 8.0. 0.5 mM EDTA and 1 mM DTT was added to the elution, and the His-TEV tag of ubiquitin was removed via TEV protease cleavage at room temperature for 4 hr. Following cleavage, a buffer exchange into 150 mM NaCl, 50 mM Tris-HCl pH 8.0 was performed using a 10 kDa centrifugal filter (Amicon). The protein solution was incubated with equilibrated cobalt resin again for 2 hr at 4°C, and UbcH7~Ub was collected in the flow-through fraction. Another centrifugal buffer exchange was performed into 20 mM NaCl, 20 mM Tris-HCl pH 7.0. The protein was passed through a HiTrap QP HP anion exchange column (GE Lifesciences) equilibrated in 20 mM NaCl, 20 mM Tris-HCl pH 7.0, and loaded onto a HiTrap SP HP cation exchange column (GE Lifesciences) in the same buffer. UbcH7~Ub was eluted using a continuous gradient of 500 mM NaCl, 20 mM Tris-HCl pH 7.0. Fractions containing UbcH7~Ub were concentrated and further purified using size-exclusion chromatography (HiLoad Superdex 16/600 S200, GE Lifesciences). The SidC-UbcH7~Ub complex was formed by incubation of SidC with UbcH7~Ub in a 1:1.2 molar ratio for 1 hr at 4°C and the complex was concentrated to a final SidC concentration of 10 mg/ml.

## Crystallization, data collection, and processing

Initial crystallization screens were carried out using a Crystal Phoenix liquid handling robot (Art Robbins Instruments) at room temperature. Crystals were grown at room temperature by hanging drop vapor diffusion by mixing 1.5 μl of protein with an equal volume of mother liquor. Small plate-like SdcA-UbcH5C crystals were formed after 2 days in 1.5 M ammonium sulfate, 3% glycerol, 0.1 M HEPES pH 7.5. The crystals were soaked in cryoprotectant solution containing the crystallization condition supplemented with 20% glycerol, and flash frozen in a stream of liquid nitrogen. Hexagonal rod-shaped SidC-UbcH7~Ub crystals formed after 4 days at room temperature in 16% PEG 3000, 0.1 M Tris pH 9.0. These crystals failed to diffract to high resolution under conventional cryo-cooling conditions. Therefore a high-pressure cryo-cooling strategy was employed to pressurize the crystals at 380 MPa for 20 min using the capillary shielding method before frozen in liquid nitrogen (*Huang et al., 2016*; *Kim et al., 2005*). Diffraction data sets were collected at MacCHESS beamline A1 at the Cornell High Energy Synchrotron Source, and indexed, integrated, and scaled with HKL-2000 (*Otwinowski and Minor, 1997*). The SdcA-UbcH5C crystal belonged to the space group $C222_1$ with unit cell dimensions a = 135.55 Å, b = 142.20 Å, c = 118.33 Å, α = β = γ = 90.0°. The SidC-UbcH7~Ub crystal belonged to the space group $P6_522$ with unit cell dimensions a = 101.52 Å,

b = 101.52 Å, c = 352.302 Å, α = β = 90.0°, γ = 120.0°. There is one complex molecule in the asymmetric unit of each of the crystals.

## Structure determination and refinement

Both structures were solved by molecular replacement using the SNL domain main lobe of the SidC542 structure (PDB ID: 4RTH) as the search model with the AMoRe program (*Trapani and Navaza, 2008*) of the CCP4 suite (*Collaborative Computational Project, 1994*). Iterative cycles of model building and refinement were performed using COOT (*Emsley and Cowtan, 2004*) and refmac5 (*Murshudov et al., 1997*) of the CCP4 suite.

## Computational analysis and graphic presentation of protein structure

Protein surface conservation was calculated by the online ConSurf server (http://consurf.tau.ac.il) (*Ashkenazy et al., 2016*). Surface hydrophobicity coloring was based on the defined hydrophobic scale (*Eisenberg et al., 1984*). Surface electrostatic potential was calculated with the APBS (*Baker et al., 2001*) plugin in PyMOL. All structural figures were generated using PyMOL (The PyMOL Molecular Graphics System, Version 1.8.X, Schrödinger, LLC). The sequences of SidC and SdcA were aligned using Clustal Omega (*Sievers et al., 2011*) and colored by the Multiple Align Show online server (http://www.bioinformatics.org/sms/index.html).

## Acknowledgements

This work is supported by the National Institute of Health grant 5R01GM116964 (YM). The X-ray data were collected at Cornell High Energy Synchrotron Source. CHESS is supported by the NSF and NIH/NIGMS via NSF award DMR-1332208, and the MacCHESS resource is supported by NIH/HIGMS award GM-103485.

## Additional information

### Funding

| Funder | Grant reference number | Author |
|---|---|---|
| National Institute of General Medical Sciences | 5R01GM116964 | Yuxin Mao |

The funders had no role in study design, data collection and interpretation, or the decision to submit the work for publication.

### Author contributions

David Jon Wasilko, Conceptualization, Formal analysis, Validation, Investigation, Writing—original draft, Writing—review and editing; Qingqiu Huang, Resources, Methodology; Yuxin Mao, Conceptualization, Formal analysis, Supervision, Funding acquisition, Validation, Writing—original draft, Project administration, Writing—review and editing

### Author ORCIDs

David Jon Wasilko http://orcid.org/0000-0002-6227-5902
Yuxin Mao http://orcid.org/0000-0002-5064-1397

### Decision letter and Author response

Decision letter https://doi.org/10.7554/eLife.36154.023
Author response https://doi.org/10.7554/eLife.36154.024

## Additional files

### Supplementary files

• Transparent reporting form

DOI: https://doi.org/10.7554/eLife.36154.017

## Data availability

Atomic coordinates and structure factors for the reported structures have been deposited into the Protein Data Bank under the accession codes 6CP0 (SdcA-UbcH5C) and 6CP2 (SidC-UbcH7~Ub).

The following datasets were generated:

| Author(s) | Year | Dataset title | Dataset URL | Database, license, and accessibility information |
|---|---|---|---|---|
| Wasilko DJ, Huang Q, Mao Y | 2018 | Structure for SdcA-UbcH5C | http://www.rcsb.org/pdb/search/structid-Search.do?structureId=6CP0 | Publicly available at the RCSB Protein Data Bank (accession no: 6CP0) |
| Wasilko DJ, Huang Q, Mao Y | 2018 | Structure for SidC-UbcH7~Ub | http://www.rcsb.org/pdb/search/structid-Search.do?structureId=6CP2 | Publicly available at the RCSB Protein Data Bank (accession no: 6CP2) |

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
