## [Decision Letter]

Thank you for submitting your article "Insights into the ubiquitin transfer cascade catalyzed by the *Legionella* effector SidC" for consideration by *eLife*. Your article has been reviewed by three peer reviewers, one of whom is a member of our Board of Reviewing Editors, and the evaluation has been overseen by Ivan Dikic as the Senior Editor. The reviewers have opted to remain anonymous.

The reviewers have discussed the reviews with one another and the Reviewing Editor has drafted this decision to help you prepare a revised submission.

Summary:

This manuscript reports two important structures of *Legionella*-encoded ubiquitin ligases, SdcA and SidC, with an E2 (SdcA-UbcH5) or with a ubiquitin-charged E2 mimetic (SidC-UbcH7-Ub), respectively. SdcA and SidC are among recently discovered ubiquitin ligases encoded by intracellular bacterial pathogens that are introduced into host cells by bacterial secretion systems to alter the biology of the host cell. SdcA/SidC are unique enzymes with no structural or sequence similarly to eukaryotic ubiquitin ligases, yet mechanistically they are similar to HECT E3s in that they function through an E3-Ub thioester intermediate. Other examples of such HECT-like bacterial E3s have been reported previously (*Salmonella* SopA, *E. coli* NleL, *Shigella* IpaH enzymes) but the *Legionella* enzymes are structurally and mechanistically distinct from these enzymes. The structures presented here yield important insight into the conformational changes that are associated with some aspects of the catalytic cycle of the SdcA/SidC enzymes and are another remarkable example of how bacterial E3s have evolved independently to mechanistically mimic their eukaryotic counterparts. This will be of high general interest to those in the ubiquitin and bacterial pathogen fields.

Essential revisions:

1) The ubiquitination assays were not well-described, making it difficult to understand the rationale for some experiments as well as to evaluate the results. The Materials and methods section should be expanded to describe both single turnover and ubiquitination experiments in greater detail, including all protein concentrations, incubation times and buffer conditions, and without referring forward to other sections. The authors should explain what is meant by use of UbcH5C as a "substrate proxy" as well as the rationale for performing the single turnover experiments in this way. Since UbcH5 and UbcH7 bind to both ligases, a concern is that the 80 µM UbcH5B may compete with UBbcH7 for binding to the E3. Further details on the assays could help clarify this issue. The authors should explain what the substrate is for "ligase activity" and full gels should be shown in supplementary data for these assays (Figure 7).

2) The authors compare structures of SdcA bound to UbcH5 with that of SidC bound to UbcH7-Ub and assume that the E3s and E2 are interchangeable. In light of the different reactivities previously reported, at least a few mutant assays should be performed with both E2s and both E3s to confirm that the mechanism is conserved and not a property of a given E2 or E3.

3) The authors previously showed that K48R mutations affect SidC ubiquitination activity, which was interpreted solely in light of chain specificity. However, the new structures show that K48 of the donor ubiquitin protrudes into a groove on the INS domain, suggesting that the behavior of K48R substitutions might be due to effects on catalysis. It would be optimal if the authors could test this directly. At the very least, they should discuss how their previous finding might be reinterpreted in light of the new structures.

4) The effect of mutations of D168, D446 and H444 are interpreted as impacting activation of the attacking substrate lysine, as observed in E2 enzymes. However, these residues are too far from the active site C46 (mutated to Ala in the structure) for this to be plausible; D168 and D446 are 12 – 14 Å from the β carbon of residue 46 and H44 is 10 Å away. These mutations must impact activity through a different mechanism, which should be considered and described. If it were possible to use the single turnover versus ubiquitin consumption assays to separate effects on transthiolation versus the aminolysis reaction, this could help in explaining the effect of these mutations.

5) The SidC structure was determined in complex with a ubiquitin-E2 conjugate where the active site Cys of the E2 was replaced with a Lys to form a stable isopeptide-linked complex. It is then stated that the active site Cys of the E3 was altered to Ala "to enhance the stability of the E3". This requires some clarification, as it is not clear why this would stabilize the E3, and since the E2-Ub complex was stable, discharge to the E3 was not a possibility.

---

## [Author Response]

Essential revisions:1) The ubiquitination assays were not well-described, making it difficult to understand the rationale for some experiments as well as to evaluate the results. The Materials and methods section should be expanded to describe both single turnover and ubiquitination experiments in greater detail, including all protein concentrations, incubation times and buffer conditions, and without referring forward to other sections. The authors should explain what is meant by use of UbcH5C as a "substrate proxy" as well as the rationale for performing the single turnover experiments in this way. Since UbcH5 and UbcH7 bind to both ligases, a concern is that the 80 µM UbcH5B may compete with UbcH7 for binding to the E3. Further details on the assays could help clarify this issue. The authors should explain what the substrate is for "ligase activity" and full gels should be shown in supplementary data for these assays (Figure 7).

We have revised the Materials and methods part describing the reaction conditions in detail. We have also provided the rationale for using UbcH5C as a substrate proxy. So far, there are no confirmed substrates that can be specifically ubiquitinated by SidC. In our in vitro ubiquitination reaction, E2s were observed to be ubiquitinated by SidC. The time-dependent accumulation of mono-ubiquitinated UbcH7 migrated at the same size as the ester-linked UbcH7~Ub on an SDS gel. Thus the turnover of ester-linked UbcH7~Ub cannot be distinguished on an SDS gel. By adding an excessive amount of UbcH5C as a substrate proxy, it can prevent/compete with SidC-mediated ubiquitination of discharged UbcH7. Moreover, since the mono-ubiquitinated UbcH5C migrated at a different size compared to the ester-linked UbcH7~Ub, the consumption of ester-linked UbcH7~Ub can be distinguished by SDS-PAGE in a time-dependent manner.

It is true that UbcH5C and UbcH7 bind to both ligases and that the 80 µM UbcH5C may compete with UbcH7 for binding to the E3, thus it may slow down the turn over reaction. However, it is still a valid approach since the same amount of UbcH5C was used in all corresponding experiments, thus the effect of UbcH5C competing for binding to E3 is likely to occur to the same extent in the assays of wild type and mutant enzymes. Moreover, the single-turn over reaction is extremely fast to be measured if wild type E2s were used. Similar to the approaches described previously (Kamadurai et al., 2009), we have used UbcH7 C86S for the preparation of ester-linked UbcH7~Ub to cripple the reaction in order to obtain measurable turnover kinetics. In this regard, adding an excessive amount of another E2 just provides an additional way to slow down the overall reaction velocity.

In the single turnover reaction of SidC, UbcH5C was used as a substrate proxy. We have compiled whole SDS gels of representative reactions in a separate supplemental figure as a related manuscript file.

2) The authors compare structures of SdcA bound to UbcH5 with that of SidC bound to UbcH7-Ub and assume that the E3s and E2 are interchangeable. In light of the different reactivities previously reported, at least a few mutant assays should be performed with both E2s and both E3s to confirm that the mechanism is conserved and not a property of a given E2 or E3.

At the SdcA-UbcH5C interface, we have created SdcA A286Y and UbcH5C F62A mutants. Both mutants showed an impaired activity in the single turnover assay (Figure 4—figure supplement 3). At the potential SdcA-Ub interface, we have generated SdcA D263R, D310R, and H500D/K501D mutants corresponding to the residues in SidC that mediate the interactions with Ub. All three mutant SdcA proteins showed defective activity by the single turnover assay (Figure 6—figure supplement 2). These results suggest that the enzymatic mechanism is conserved in both SidC and SdcA.

3) The authors previously showed that K48R mutations affect SidC ubiquitination activity, which was interpreted solely in light of chain specificity. However, the new structures show that K48 of the donor ubiquitin protrudes into a groove on the INS domain, suggesting that the behavior of K48R substitutions might be due to effects on catalysis. It would be optimal if the authors could test this directly. At the very least, they should discuss how their previous finding might be reinterpreted in light of the new structures.

In our previous publication, we observed a reduced activity of poly-Ub chain formation by Ub48K, in which all lysine residues are mutated to arginine except for K48 (Hsu et al., 2014; Figure 4B). However, SidC ligase activity with the single K48R mutant was not significantly different compared to wild type Ub (Hsu et al., 2014; Figure 4A). These data suggested that SidC can catalyze the formation of K48 linked Ub chain, although the K48 chain is not preferred. The single K48R mutation has no significant impact on the ligase activity, suggesting that the interaction between K48 and the INS domain is not critical to the ligase activity.

4) The effect of mutations of D168, D446 and H444 are interpreted as impacting activation of the attacking substrate lysine, as observed in E2 enzymes. However, these residues are too far from the active site C46 (mutated to Ala in the structure) for this to be plausible; D168 and D446 are 12 – 14 Å from the β carbon of residue 46 and H44 is 10 Å away. These mutations must impact activity through a different mechanism, which should be considered and described. If it were possible to use the single turnover versus ubiquitin consumption assays to separate effects on transthiolation versus the aminolysis reaction, this could help in explaining the effect of these mutations.

Indeed, in the SidC-UbcH7~Ub ternary complex, the distances between the catalytic residue C46 with D168, D446, H444 are 12-14 Å apart. However, the loop containing the catalytic cysteine is flexible, as evidenced in the SdcA-UbcH5C structure, in which these distances are closer to 5 Å. A potential conformational change would allow for the catalytic cysteine to move in close proximity to the cluster of acidic residues that might function to activate the incoming substrate lysine. We have added a figure to demonstrate the conformational flexibility of the active site (Figure 7—figure supplement 1).

5) The SidC structure was determined in complex with a ubiquitin-E2 conjugate where the active site Cys of the E2 was replaced with a Lys to form a stable isopeptide-linked complex. It is then stated that the active site Cys of the E3 was altered to Ala "to enhance the stability of the E3". This requires some clarification, as it is not clear why this would stabilize the E3, and since the E2-Ub complex was stable, discharge to the E3 was not a possibility.

We found that the SidC C46A construct expressed at a higher level and showed fewer impurities upon purification. We reasoned that this construct was more stable and used the catalytic mutant SidC/SdcA proteins for the crystallographic experiments.